# A Nonlinear Adaptive Autopilot for Unmanned Aerial Vehicles Based on the Extension of Regression Matrix

**Quanwen Hu** [†]**, Yue Feng** [†]**, Liaoni Wu** *,[†] and **Bin Xi** [†]

School of Aerospace Engineering, Xiamen University, Xiamen 361102, China
* Correspondence: wuliaoni@xmu.edu.cn; Tel.: +86-1815-035-6682
† These authors contributed equally to this work.

**Abstract:** In applications of the $L_1$ adaptive flight control system, we found two limitations to be extended: (1) the system cannot meet the demands of engineering in terms of nonlinearity and adaptation in most flight scenarios; (2) the adaptive control law generates a transient response in the tracking error, hindering the system from reaching the steady-state error, and ultimately decreasing control accuracy. In response to these problems, an extended flight control system for $L_1$ adaptive theory is proposed and rigorously proved. This system involves considering the nonlinear function matrix of state variables, which serves as an extension of the regression matrix in the original $L_1$ adaptive control system, thus enhancing its nonlinear characteristics. The problem of calculating the adaptive laws, caused by the extended regression matrix, is solved by using the pseudo-inverse matrix. To eliminate the transient response, the state vector and its estimate are recorded and employed just like an integrator. Finally, the proposed system is verified on a high-subsonic flight subject to nonlinear uncertainties, with simulation results showing improved control accuracy and enhanced robustness. The proposed system resolves the limitations of the $L_1$ adaptive control system in nonlinearity, providing the possibility for further theoretical development to improve the performance of adaptive control systems.

**Keywords:** nonlinear control; adaptive control; drone development; pseudo-inverse matrix; Monte Carlo



## 1. Introduction

Unmanned aerial vehicles (UAV) are controlled by computational flight control instead of human pilots. The autopilot is the most critical piece of equipment for UAVs to achieve autonomous flight, and the controller within the autopilot serves as the core for its proper functioning. Early laws of the controller for autopilots usually involved the use of linear time-invariant (LTI) controllers. These controllers often exhibit poor performance in the presence of nonlinear uncertainties, and the autopilots utilizing such controllers are not adaptive. Adaptive autopilots can enhance the flight control and navigation capabilities required for autonomous flight. They can also adjust to changes in flight conditions, such as wind gusts or turbulence, and make real-time adjustments to the UAV's flight path to maintain stability, accuracy, etc. In recent years, the requirements for reliability, comfort, and maneuverability of new-generation UAVs have raised the bar for control accuracy, control response, and control stability of autopilot controllers. It is still necessary to develop adaptive autopilots and controllers within them [1]. The controller of new-generation adaptive autopilots has attracted extensive interest among researchers, and numerous control theories have been proposed. Various effective control methods have been applied to control UAV flight, which have delivered good performance. Mohammad utilized model reference adaptive control (MRAC) to track the attitude of medium-scale UAVs [2]. Karim Ahmadi conducted applications of nonlinear dynamic inversion control to quad-rotors [3], and L. A. Blas achieved similar applications with active disturbance

rejection control [4]. JR Montoya-Morales applied sliding mode variable structure control to solve an autonomous trajectory-tracking problem [5]. Wenbo Gao employed deep learning control in the aeroengine control system and significantly improved the performance of this engine [6]. Guerrero-Sánchez presented a passivity-based control scheme and enhanced the performance of UAVs in load transportation [7,8]. Among all theories, the MRAC controller, offering high control accuracy, robustness, and adaptability [9–13], is one of the most mature and widely used adaptive controllers. As an extension of MRAC, the $L_1$ adaptive controller preserves these advantages while introducing novel features. It is worth conducting more in-depth research on this controller.

Unlike MRAC or other adaptive control theories, $L_1$ adaptive control does not require accurate knowledge of system parameters or model structures. Instead, it uses a novel "fast adaptation" mechanism that rapidly estimates system parameters and adapts the control law accordingly. The key advantage of $L_1$ adaptive control is that it provides high robustness to parameter uncertainty and disturbances, while maintaining good tracking performance. The abbreviation "$L_1$" precisely highlights this key advantage, as it stands for performance in terms of attenuating external disturbances while maintaining or improving the system's response to internal signals. $L_1$ adaptive control is usually used to solve control system problems that are nonlinear, time-varying, multivariable, and uncertain. Overall, this controller is highly suitable for the control systems of new-generation UAVs.

An $L_1$ adaptive controller consists of a state predictor, laws of adaptation, and a control law. In the context of differences in the laws of adaptation, $L_1$ adaptive control can be divided into $L_1$ adaptive control based on the Lyapunov stability theorem ($L_1$-Ly) and control based on the piecewise constant laws of adaptation ($L_1$-Pc). $L_1$-Ly provides a convenient theoretical approach for the proof of stability and the analysis of control performance and has been subjected to greater research and generated more applications [14–20] than $L_1$-Pc. However, minor changes in the parameters can result in significant changes in the performance of the $L_1$-Ly controller, and this poses challenges in terms of adjusting its parameters. The $L_1$-Pc controller is more suitable for engineering applications than the $L_1$-Ly controller because it has fewer parameters and thus is easier to use. Further, this controller can achieve high adaptive speeds through improvements in the performance of the hardware. We thus investigate the $L_1$-Pc controller here.

In recent years, a growing number of scholars have devoted themselves to theoretical research [21–24], the evaluation [25–27] and the application [28–32] of $L_1$-Pc. Jintasit [29] combined model-predictive path integral control with $L_1$ adaptive control to achieve effective trajectory tracking for quad-rotor aircraft. However, high-frequency oscillations in the body rate were observed in some cases in the literature. Hanover [31] further optimized this theory, but the problem remained unsolved. We suspected that these oscillations might be due to the inadequate adaptability of the $L_1$ adaptive controller. Additionally, we observed deviations between the actual and reference states in several papers [24,30,31]. Thus, we examined the L1 controller and eventually identified that the $L_1$-Pc control system still has two limitations that remain unresolved. On the one hand, $L_1$-Pc can be regarded as an LTI control system in most flight scenarios (as proved in Section 3), but cannot meet the demands of engineering in terms of nonlinearity and adaptation. On the other hand, $L_1$-Pc controllers generate transient responses that always exist. This inference can be obtained by the following reasoning: If the state predictor of the $L_1$-Pc controller works perfectly, the estimate of a state vector (or an uncertainty) should be the same as its actual value. In addition, the original laws of adaptation show that the estimate of uncertainty can be regarded as a linear combination of deviations between the actual state vector and its estimate. Thus, when the actual state vector is equal to its estimate in a steady state of control response, the estimate of uncertainty should be zero. Then, the uncertainty of the object of control should also be zero, which is impossible for actual flight. Therefore, either steady-state deviations or a dynamically stable state must exist, and both result in poor performance of $L_1$-Pc controllers.

To optimize the $L_1$-Pc controller, we made numerous revisions to it and finally proposed an adaptive flight control system with the extension of a regression matrix ($L_1$-PcEx). To prevent an $L_1$-Pc controller from becoming an LTI controller, we extended the regression matrix in the laws of adaptation, i.e., we replaced a constant with a column vector. The column vector is a nonlinear function matrix of state variables, and the entries in it consist of "1" and the time-variant state variables of the object of control. In this way, we enhanced the nonlinearity of the original controller. We also used the pseudo-inverse matrix [33], so that all matrices can perform a division-like operation to calculate the laws of adaptation. Furthermore, to avoid unexpected transient responses, a cumulative value of the state errors between the state vector and its estimate was applied to the laws of adaptation like an integrator. In this way, the $L_1$-PcEx controller can gradually remove the unexpected transient response. In general, we modified the framework of the $L_1$-Pc control system in accordance with the laws of adaptation, which enhances the safety and comfort of automatic cruise piloting of civil airliners. This, in turn, improves the reputation of the civil aviation administration and customer satisfaction. $L_1$-PcEx controllers can also ensure that high-speed UAVs deliver good performance, even under extreme conditions, and thus can reliably be applied to military scenarios including defense reconnaissance, air combat, and extreme escape.

The remainder of this paper is organized as follows: Section 2 introduces a model of a high-speed UAV, and Section 3 discusses the control framework of $L_1$-PcEx. Section 4 reports a performance analysis of $L_1$-PcEx, consisting mainly of the analysis of the reference system, transient performance, and steady-state performance. Section 5 presents the results of simulations based on high-subsonic UAVs, including the results of flight feasibility simulations and algorithmic comparison, as well as Monte Carlo simulations. Section 6 presents the conclusion of this study.

## 2. Problem Formulation

Mathematical models are the basis for the analysis of the characteristics of a UAV, the design of its control system, and its simulation. The model of an UAV is usually expressed as a 12th-order equation with six degrees of freedom, containing triaxial speeds, angles, angular rates, and positions.

In this section, we will introduce the general state-space equation form for a controlled system, which is applicable to a wide range of controlled objects, including UAVs. This equation form is commonly used in control engineering to model and design control systems and can be tailored to meet the specific requirements and characteristics of the controlled object. A nonlinear model of a UAV can be linearized around an equilibrium state using small-perturbation theory [34]. The linearized model of a UAV can easily be expressed in the form of state-space equations. The transformation of the 12th-order equation into a state-space equation has been provided in Refs. [35,36].

Consider the following multiple-in multiple-out (MIMO) system:

$$
\begin{aligned}
&\dot{x}(t) = A_m x(t) + B_m u(t) + f(t, x(t), z(t)),\ x(0) = x_0,\\
&\dot{x}_z(t) = g(t, x_z(t), x(t)),\ x_z(0) = x_{z0},\\
&z(t) = g_0(t, x_z(t)),\\
&y(t) = C x(t),
\end{aligned}
\tag{1}
$$

where $x(t) \in \mathbb{R}^n$ is the state vector of this system; $u(t) \in \mathbb{R}^m$ is the control signal($m \leq n$); $y(t) \in \mathbb{R}^m$ is the regulated output; $A_m \in \mathbb{R}^{n \times n}$ is a known Hurwitz matrix that defines the desired dynamics of a closed-loop system; $B_m \in \mathbb{R}^{n \times m}$ is a known full-rank constant matrix [$(A_m, B_m)$ is controllable]; $C \in \mathbb{R}^{m \times n}$ is a known full-rank constant matrix [$(A_m, C)$ is observable]; $z(t) \in \mathbb{R}^p$ is the output vector of internal unmodeled dynamics; $x_z(t) \in \mathbb{R}^l$ is the state vector of internal unmodeled dynamics; and $f : \mathbb{R} \times \mathbb{R}^n \times \mathbb{R}^p \to \mathbb{R}^n$, $g_0 : \mathbb{R} \times \mathbb{R}^l \to \mathbb{R}^p$, and $g : \mathbb{R} \times \mathbb{R}^l \times \mathbb{R}^n \to \mathbb{R}^l$ are unknown nonlinear functions satisfying the

standard assumptions of existence and uniqueness. The initial condition $x_0$ is assumed to be inside an arbitrarily large known set, i.e., $\|x_0\|_\infty \leq \rho_0 < \infty$ for $\rho > 0$.

For subsequent analyses, the model of the UAV can be rewritten in the following form:

$$
\begin{aligned}
\dot{x}(t) &= A_m x(t) + B_m[u(t) + f_1(t, x(t), z(t))] + B_{um} f_2(t, x(t), z(t)), \\
\dot{x}_z(t) &= g_0(t, x_z(t), x(t)), \; x_z(0) = x_{z0}, \\
x(0) &= x_0, \; x_z(0) = x_{z0}, \\
z(t) &= g_0(t, x_z(t)), \\
y(t) &= Cx(t),
\end{aligned}
\tag{2}
$$

where $B_m \in \mathbb{R}^{n \times (n-m)}$ is a constant matrix, such that $B_m^T \cdot B_{um} = 0$, $rank[B_m, B_{um}] = n$, $f_1 : \mathbb{R} \times \mathbb{R}^n \times \mathbb{R}^p \to \mathbb{R}^m$, and $f_2 : \mathbb{R} \times \mathbb{R}^n \times \mathbb{R}^p \to \mathbb{R}^{(n-m)}$ are unknown nonlinear functions that satisfy:

$$
\begin{bmatrix} f_1(t, x(t), z(t)) \\ f_2(t, x(t), z(t)) \end{bmatrix} = B^{-1} f(t, x(t), z(t)), \; B = [B_m, B_{um}].
\tag{3}
$$

Furthermore, because of the low level of coupling between the longitudinal and the lateral flight modes [35,36] it is reasonable to study them separately. This paper examines the longitudinal motion of high-subsonic UAVs. Because changes in the pitch angle and the pitch rate greatly affect flight performance, the state vector we are interested in in this article is $x = [\vartheta, Q]^T$, and the input value we are interested in is $u = \delta_E$. $\vartheta$ is the pitch angle of the UAV, $Q$ is its pitch rate, and $\delta_E$ is the elevator.

## 3. $L_1$-PcEx Architecture

The $L_1$-PcEx architecture with its main elements is shown in Figure 1.

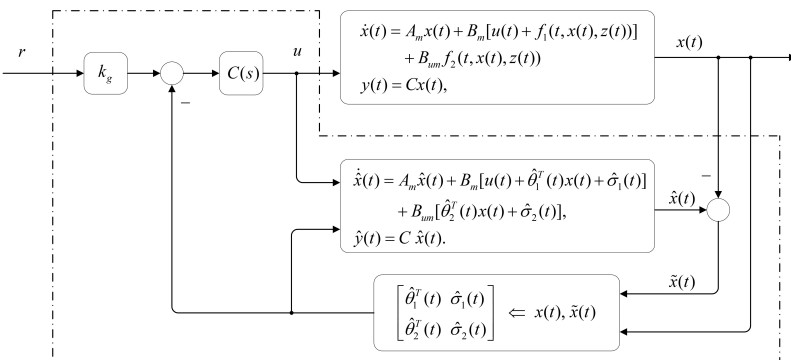

**Figure 1.** Closed-loop $L_1$-PcEx system.

Following Ref. [14], let $H(s) = B_m(sI_m - Am)^{-1}$ and let $C(s)$ be a filter. Thus we have Equation (4), implying that the reference system of $L_1$-Pc is equivalent to that of an LTI controller:

$$
u_{ref}(s) = \frac{C(s)}{1 - C(s)} K_g r(s) - \frac{C(s)}{1 - C(s)H(s)} x_{ref}(s).
\tag{4}
$$

Moreover, by using Ref. [21] with a given adaptation sampling time $T_s > 0$, we can arrive at the following conclusions for a $L_1$-Pc control system:

$$
\Delta\hat{\sigma} = \begin{bmatrix} \Delta\hat{\sigma}_1 \\ \Delta\hat{\sigma}_2 \end{bmatrix} = -\Phi^{-1}(T_s)e^{A_m T_s}\Delta\hat{x} + \Phi^{-1}(T_s)e^{A_m T_s}\Delta x,
\tag{5}
$$

where $\Phi(T_s) = \int_0^{T_s} e^{A_m(T_s-\tau)} B d\tau$, $\Delta x = x((k+1)T_s) - x(kT_s)$, $\Delta \hat{x} = \hat{x}((k+1)T_s) - \hat{x}(kT_s)$, and $\Delta \hat{\sigma} = \hat{\sigma}((k+1)T_s) - \hat{\sigma}(iT_s)$. The variable $k$ represents the index of the moment, while $x(kT_s)$ represents the value of $x$ at the $kT_s$-th discrete time point.

Let $\Delta \sigma = F_{x|x_0} \Delta x + F_{\hat{x}|\hat{x}_0} \Delta \hat{x}$, where $F_{x|x_0} = \Phi^{-1}(T_s) e^{A_m T_s}$ and $F_{\hat{x}|\hat{x}_0} = -\Phi^{-1}(T_s) e^{A_m T_s}$. Then, using Ref. [14]:

$$\Delta u(s) = K_{\Delta x}(s) \Delta x(s) = C(s)[1 - F_{\hat{x}|\hat{x}_0} H(s)(1 - C(s))]^{-1} F_{x|x_0} \Delta x(s). \tag{6}$$

Further, we have:

$$\lim_{T_s \to 0} K_{\Delta x}(s) = \lim_{T_s \to 0} \frac{C(s)\Phi^{-1}(T_s)e^{A_m T_s}}{1 + \Phi^{-1}(T_s)e^{A_m T_s} H(s)(1 - C(s))} = \frac{C(s)}{H(s)(1 - C(s))}. \tag{7}$$

With the gain $K_{\Delta x}(s)$ in Equation (6), we conclude that $L_1$-Pc is an LTI system. When $T_s$ is small enough (Equation (7)), the performance of $L_1$-Pc depends solely on the $L_1$ filter and H(s), and is not affected by nonlinearity or initial conditions.

To enable the controller to adaptively adjust to the influence of nonlinearity and initial conditions, we designed the L1-PcEx controller. The architecture of $L_1$-PcEx is defined as below:

**State Predictor**

Consider the following state predictor:

$$\dot{\hat{x}}(t) = A_m \hat{x}(t) + B_m[u(t) + \hat{\theta}_1^T(t)x(t) + \hat{\sigma}_1(t)] + B_{um}[\hat{\theta}_2^T(t)x(t) + \hat{\sigma}_2(t)] \tag{8}$$
$$\hat{y}(t) = C\hat{x}(t),$$

where $\hat{\theta}_1^T(t)$, $\hat{\theta}_2^T(t) \in \mathbb{R}^{m \times n}$ and $\hat{\sigma}_1(t)$, $\hat{\sigma}_2(t) \in \mathbb{R}^m$ are both adaptive estimates.

**Adaptation laws**

The laws of adaptation for $\hat{\theta}_1^T(t)$, $\hat{\theta}_2^T(t)$, $\hat{\sigma}_1(t)$, $\hat{\sigma}_2(t)$ are defined as:

$$\begin{bmatrix} \hat{\theta}_1^T(t) & \hat{\sigma}_1(t) \\ \hat{\theta}_2^T(t) & \hat{\sigma}_2(t) \end{bmatrix} = \begin{bmatrix} \hat{\theta}_1^T(iT_s) & \hat{\sigma}_1(iT_s) \\ \hat{\theta}_2^T(iT_s) & \hat{\sigma}_2(iT_s) \end{bmatrix}, t \in [iT_s, (i+1)T_s],$$
$$\begin{bmatrix} \hat{\theta}_1^T(iT_s) & \hat{\sigma}_1(iT_s) \\ \hat{\theta}_2^T(iT_s) & \hat{\sigma}_2(iT_s) \end{bmatrix} = [-\Phi^{-1}(T_s)e^{A_m T_s}\tilde{x}(iT_s) + \Phi^{-1}h(iT_s)][\Psi(iT_s)]^+, \tag{9}$$
$$h(iT_s) = -\tilde{x}(iT_s) + h[(i-1)T_s], h(0) = 0,$$

where $\Phi(t) = \int_0^t e^{A_m(t-\tau)} B d\tau$, $\Psi(t) = [x(t), 1]^T$, $[\Psi(t)]^+$ is the pseudo-inverse matrix of $\Psi(t)$, the variable $i$ is the index of the moment, and $T_s > 0$ is the sampling time for adaptation.

**Control Law**

The control signals are generated as outputs of the system:

$$u(s) = -KD(s)[u(s) + \hat{\eta}_1(s) + \hat{\eta}_{2m}(s) - r_g(s)], \tag{10}$$

where

$$\hat{\eta}_1(s) = [\hat{\theta}_1^T(t), \hat{\sigma}_1(s)] \cdot \Psi(s),$$
$$\hat{\eta}_{2m}(s) = H_m^{-1}(s)H_{um}(s)\hat{\eta}_2(s),$$
$$H_m(s) = C(sI_n - A_m)^{-1}B_m,$$
$$H_{um}(s) = D(sI_n - A_m)^{-1}B_{um}, \tag{11}$$
$$\hat{\eta}_2(s) = [\hat{\theta}_2^T(t), \hat{\sigma}_2(s)] \cdot \Psi(s),$$
$$r_g = k_g(s)r(s).$$

## 4. Analysis of the $L_1$-PcEx

### 4.1. Assumptions and Definitions

The model given in Section 2 is extremely complex. For the sake of argument, the following four assumptions are made. Assumption 1 ensures that the $f(t, x(t), z(t))$ is bounded when $x(t, z(t)) = 0$; Assumption 2 is the semiglobal Lipschitz condition; Assumption 3 ensures the stability of unmodeled dynamics; Assumption 4 ensures the stability of matched transmission zero. The system in Section 2 verifies those assumptions. All results are based on them.

**Assumption 1.** *For all $t \geq 0$, there exists $B_1, B_2 > 0$, such that $\|f_1(t, 0)\|_\infty \leq B_1$ and $\|f_2(t, 0)\|_\infty \leq B_2$ hold.*

**Assumption 2.** *Define $X_1 = [x_1, z_1]^T$, $X_2 = [x_2, z_2]^T$. For arbitrary $\delta > 0$, for $\|X_1\|_\infty$, $\|X_2\|_\infty \leq \delta$, there exists positive $K_{i\delta}(i = 1, 2)$, such that $\|f_i(t, X_1) - f_i(t, X_2)\|_\infty \leq K_{i\delta}\|X_1 - X_2\|_\infty$ uniformly holds.*

**Assumption 3.** *With respect to both initial conditions $x_{z0}$ and input $x(t)$, the $x_z$-dynamics are bounded input and bounded output (BIBO) stable, i.e., for all $t \geq 0$, there exists $L_z, B_z > 0$, such that the following equation holds: $\|z_t\|_\infty \leq L_z\|x_t\|_\infty + B_z$.*

**Assumption 4.** *The transmission zeros of the transfer matrix $H_m(s) = C(sI_n - A_m)^{-1}B_m$ lie in the open left half-plane.*

For convenience, let:

$$
\begin{aligned}
H_{xm}(s) &= (sI_n - A_m)^{-1}B_m, \\
H_{xum}(s) &= (sI_n - A_m)^{-1}B_m, \\
H_m(s) &= CH_{xm}(s) = (sI_n - A_m)^{-1}B_m, \\
H_{um}(s) &= CH_{xum}(s) = (sI_n - A_m)^{-1}B_{um}.
\end{aligned}
\tag{12}
$$

Let $x_{in}(t)$ be the signal with Laplace transform $x_{in}(t) = (sI_n - A_m)^{-1}x_0$. If $\rho_{in} = \|(sI_n - A_m)^{-1}\|_{L_1}\rho_0$, then $\|x_{in}\|_{L_\infty} \leq \rho_{in}$

For every $\delta > 0$, let

$$
L_{i\delta} = \bar{\delta}(\delta)K_{i\bar{\delta}}/\delta, \quad \bar{\delta}(\delta) = max\{\delta + \bar{\gamma}_1, L_z(\delta + \bar{\gamma}_1) + B_z\}
\tag{13}
$$

where $i = 1, 2$ and $\bar{\gamma}_1$ is defined as follows:

$$
\bar{\gamma}_1 = \frac{\|H_{xm}(s)C(s)H_m^{-1}(s)C\|_{L_1}}{1 - \|G_m(s)\|_{L_1}L_{1\rho_r} - \|G_{um}(s)\|_{L_1}L_{2\rho_r}}\bar{\gamma}_0 + \beta
\tag{14}
$$

where both $\bar{\gamma}_0$ and $\beta$ are arbitrarily small positive constants and $\gamma_1 \leq \bar{\gamma}_1$.

The design of the $L_1$-PcEx controller involves a feedback gain matrix $K \in \mathbb{R}^{m \times m}$ and a strictly proper transfer matrix $D(s)$, leading to a strictly proper filter:

$$
C(s) = (I_m + KD(s))^{-1}KD(s),
\tag{15}
$$

with $C(0) = I_m$. The choice of $D(s)$ ensures that $C(s)H_m^{-1}(s)$ is a proper stable transfer matrix.

For the proofs of stability and performance bounds, the choices for $K$ and $D(s)$ must ensure that, for a given $\rho_0$, there exists $\rho_r > \rho_{in}$, such that the following $L_1$-norm condition holds:

$$\|G_m(s)\|_{L_1} + \|G_{um}(s)\|_{L_1} l_0 < \frac{\rho_r - \|H_{xm}(s)C(s)K_g(s)\|_{L_1}\|r\|_{L_\infty} - \rho_{in}}{L_{1\rho_r}\rho_r + B_0}, \tag{16}$$

where

$$
\begin{aligned}
G_m(s) &= H_{xm}(s)(I_m - C(s)), \\
G_{um}(s) &= [I_n - H_{xm}(s)C(s)H_m^{-1}(s)C]H_{xum}(s), \\
l_0 &= L_{2\rho_r}/L_{1\rho_r}, \ B_0 = max\{B_{10}, B_{20}\}.
\end{aligned} \tag{17}
$$

$K_g(s)$ is a BIBO-stable feedforward prefilter. Notably, Equation (16) is a prerequisite for the stability and good performance of $L_1$-PcEx.

Let

$$
\begin{aligned}
\rho &= \rho_r + \overline{\gamma_1}, \\
\rho_u &= \rho_{ur} + \gamma_2,
\end{aligned} \tag{18}
$$

where

$$
\begin{aligned}
\rho_{ur} &= \|C(s)\|_{L_1}(L_{1\rho_r}\rho_r + B_{10}) + \|C(s)H_m^{-1}(s)H_{um}(s)\|_{L_1}(L_{2\rho_r}\rho_r + B_{20}) \\
&\quad + \|C(s)K_g(s)\|_{L_1}\|r\|_{L_\infty}, \\
\gamma_2 &= \|C(s)\|_{L_1}L_{1\rho_r}\gamma_1 + \|C(s)H_m^{-1}(s)H_{um}(s)\|_{L_1}L_{2\rho_r}\gamma_1 + \|C(s)H_m^{-1}(s)C\|_{L_1}\overline{\gamma_0}.
\end{aligned} \tag{19}
$$

Moreover, let $T_s > 0$ be the sampling time for adaptation, which is associated with the sampling rate of the available CPU. Let $\zeta(T_s)$ be:

$$\zeta(T_s) = \kappa_1(T_s)\Delta_1 + \kappa_2(T_s)\Delta_2, \tag{20}$$

where $\kappa_1(T_s)$ and $\kappa_2(T_s)$ are defined as:

$$
\begin{aligned}
\kappa_1(T_s) &= \int_0^{T_s} \|e^{A_m(T_s-\tau)}B_m\|_2 d\tau, \\
\kappa_2(T_s) &= \int_0^{T_s} \|e^{A_m(T_s-\tau)}B_{um}\|_2 d\tau,
\end{aligned} \tag{21}
$$

and $\Delta_1$, $\Delta_2$ are given by:

$$
\begin{aligned}
\Delta_1 &= (L_{m\rho}\rho + B_{m0})\sqrt{m}, \\
\Delta_2 &= (L_{um\rho}\rho + B_{um0})\sqrt{n-m}.
\end{aligned} \tag{22}
$$

Let $\alpha_1(t)$, $\alpha_2(t)$, $\alpha_3(t)$, $\alpha_4(t)$ be defined as:

$$
\begin{aligned}
\alpha_1(t) &= \|e^{A_m t}\|_2, \\
\alpha_2(t) &= \int_0^t \|e^{A_m(t-\tau)}\Phi^{-1}e^{A_m T_s}\|_2 d\tau, \\
\alpha_3(t) &= \int_0^t \|e^{A_m(t-\tau)}B_m\|_2 d\tau, \\
\alpha_4(t) &= \int_0^t \|e^{A_m(t-\tau)}B_{um}\|_2 d\tau.
\end{aligned} \tag{23}
$$

where $\Phi(t)$ is a square matrix of order n defined as:

$$\Phi(t) = \int_0^t e^{A_m(t-\tau)} B d\tau. \tag{24}$$

Moreover, let $\bar{\alpha}_1(T_s)$, $\bar{\alpha}_2(T_s)$, $\bar{\alpha}_3(T_s)$, $\bar{\alpha}_4(T_s)$ be defined as:

$$\begin{aligned}
\bar{\alpha}_1(T_s) &= \max_{t \in [0, T_s]} \alpha_1(t), \quad \bar{\alpha}_2(T_s) = \max_{t \in [0, T_s]} \alpha_2(t), \\
\bar{\alpha}_3(T_s) &= \max_{t \in [0, T_s]} \alpha_3(t), \quad \bar{\alpha}_4(T_s) = \max_{t \in [0, T_s]} \alpha_4(t),
\end{aligned} \tag{25}$$

Finally, let

$$\gamma_0(T_s) = [2\bar{\alpha}_1(T_s) + 2\bar{\alpha}_2(T_s) + 1]\zeta(T_s) + \bar{\alpha}_3(T_s)\Delta_1 + \bar{\alpha}_4(T_s)\Delta_2. \tag{26}$$

*4.2. Closed-Loop Reference System*

Consider a closed-loop reference system:

$$\begin{aligned}
\dot{x}_{ref}(t) &= A_m x_{ref}(t) + B_m[u(t) + f_1(t, x_{ref}(t), z(t))] + B_{um} f_2(t, x_{ref}(t), z(t)), \\
u_{ref}(t) &= -C(s)[\eta_{1ref}(s) + H_m^{-1}(s)H_{um}(s)]\eta_{2ref}(s) - K_g(s)r(s)], \\
y_{ref}(t) &= C x_{ref}(t),
\end{aligned} \tag{27}$$

where $\eta_{iref}(s)$ is the Laplace transform of $\eta_{iref}(t) = f_i(t, x_{ref}(t), z(t))$, for $i = 1, 2$.

**Lemma 1.** *For a closed-loop reference system that satisfies Equation (16), if $\|x_0\|_\infty \leq \rho_0$ and $\|z_t\|_{L_\infty} \leq L_z(\|x_{ref\ t}\|_{L_\infty} + \gamma_1) + B_z$, then the following conclusion can be drawn:*

$$\|x_{ref\ t}\|_{L_\infty} \leq \rho_r, \tag{28}$$

$$\|u_{ref\ t}\|_{L_\infty} \leq \rho_{ur}, \tag{29}$$

**Proof.** For the closed-loop reference system at $\forall t \in [0, \tau]$, we have:

$$x_{ref} = G_m(s)\eta_{1ref}(s) + G_{um}(s)\eta_{2ref}(s) + H_{xm}(s)C(s)K_g(s)r(s) + x_{in}(s). \tag{30}$$

□

Using the definition and properties of a vector norm and matrix norm, the following conclusions can be obtained:

$$\begin{aligned}
\|x_{ref}\|_{L_\infty} = &\|G_m(s)\|_{L_1}\|\eta_{1ref}(s)\|_{L_\infty} + \|G_{um}(s)\|_{L_1}\|\eta_{2ref}(s)\|_{L_\infty} \\
&+ \|H_{xm}(s)C(s)K_g(s)\|_{L_1}\|r(s)\|_{L_\infty} + \rho_{in}.
\end{aligned} \tag{31}$$

If $\|x_{ref}(t)\|_\infty \leq \rho_r$ is not true, since $\|x_{ref}(0)\|_\infty = \|x_0\|_\infty \leq \rho_r$ and $x_{ref}(t)$ are continuous, there must exist a time $\tau_1 \in (0, \tau]$, such that:

$$\begin{aligned}
\|x_{ref}(t)\|_\infty &< \rho_r, \ \forall t \in [0, \tau_1), \\
\|x_{ref}(\tau_1)\|_\infty &= \rho_r.
\end{aligned} \tag{32}$$

In another words,

$$\|x_{ref\ \tau_1}\|_{L_\infty} = \rho_r. \tag{33}$$

According to Equation (28),

$$\|z_{\tau_1}\|_{L_\infty} \leq Lz(\rho_r + \gamma_1) + B_z, \tag{34}$$

and hence, we have:

$$\|X_{ref\ \tau_1}\|_{L_\infty} = \begin{bmatrix} x_{ref}^T & z^T \end{bmatrix} \leq \overline{\rho}_r(\rho_r) = max\{\rho_r + \gamma_1, \ L_z(\rho_r + \gamma_1) + B_z\}. \tag{35}$$

Then, it follows from Assumptions 1 and 2 that:

$$\begin{aligned}
\|\eta_{iref\ \tau_1}\|_{L_\infty} &\leq K_{i\overline{\rho}_r(\overline{\rho}_r)}\|X_{ref\ \tau_1}\|_{L_\infty} + B_{i0} \leq K_{i\overline{\rho}_r}(\overline{\rho}_r)\overline{\rho}_r + B_{i0}, \\
\|\eta_{iref\ \tau_1}\|_{L_\infty} &\leq L_{i\rho_r}\rho_r + B_{i0},
\end{aligned} \tag{36}$$

for $i = 1,\ 2$.

Thus, at moment $\tau_1$, Equation (31) can be rewritten as:

$$\begin{aligned}
\|x_{ref\ \tau_1}\|_{L_\infty} &\leq \|G_m(s)\|_{L_1}(L_{1\rho_r}\rho_r + B_{10}) + \|G_{um}(s)\|_{L_1}(L_{2\rho_r}\rho_r + B_{20}) \\
&\quad + \|H_{xm}(s)C(s)K_g(s)\|_{L_1}\|r(s)\|_{L_\infty} + \rho_{in},
\end{aligned} \tag{37}$$

and with Equation (16), we have:

$$\|x_{ref\ \tau_1}\|_{L_\infty} \leq \rho_r. \tag{38}$$

Consequently, Equation (38) contradicts Equation (33). Therefore, $\|x_{ref\ t}\|_{L_\infty} \leq \rho_r$ is proved, and for all $t \in (0,\ \tau]$, we have

$$\|\eta_{iref\ \tau}\|_{L_\infty} \leq L_{i\rho_r}\rho_r + B_{i0}, \ i = 1, 2. \tag{39}$$

Finally, using Equations (27) and (38), we have:

$$\|u_{ref\ \tau}\|_{L_\infty} \leq \rho_{ur}. \tag{40}$$

That proves Equation (29).

**Remark 1.** *The boundedness of the reference system ensures that the target of the object of control is stable, and this forms a solid foundation for good control performance.*

*4.3. Transient and Steady-State Performance*

From the object of control and the reference system, the error dynamics can be derived as:

$$\dot{\tilde{x}}(t) = A_m\tilde{x}(t) + B_m\tilde{\eta}_1(t) + B_{um}\tilde{\eta}_2, \tag{41}$$

where

$$\begin{aligned}
\tilde{\eta}_1(t) &= \hat{\theta}_1^T(t)x(t) + \hat{\sigma}_1(t) - \eta_1(t), \\
\tilde{\eta}_2(t) &= \hat{\theta}_2^T(t)x(t) + \hat{\sigma}_2(t) - \eta_2(t).
\end{aligned} \tag{42}$$

**Lemma 2.** *If the controller is subject to the $L_1$-norm condition (Equation (16)), $\|x_\tau\|_{L_\infty} \leq \rho$ and $\|u_\tau\|_{L_\infty} \leq \rho_u$, we have*

$$\|\tilde{x}_\tau\|_{L_\infty} \leq \overline{\gamma}_0, \tag{43}$$

*where $\gamma_0 \leq \overline{\gamma}_0$.*

**Proof.** According to Assumption 3, we then have:

$$\|X_\tau\|_{L_\infty} \leq \overline{\rho}(\rho) = max\{\rho, \ L_z\rho + B_z\}. \tag{44}$$

$\square$

From Assumptions 1 and 2, we obtain:

$$\|\eta_{i\tau}\|_{L_\infty} \le L_{i\rho}\rho + B_{i0}, \ i = 1, 2. \tag{45}$$

Then, according to the definitions of $\|\cdot\|_{L_\infty}$ and $\|\cdot\|_2$, we have:

$$
\begin{aligned}
\|\eta_1(t)\|_2 &\le (L_{1\rho}\rho + B_{10})\sqrt{m}, \forall t \in [0, \tau], \\
\|\eta_2(t)\|_2 &\le (L_{2\rho}\rho + B_{20})\sqrt{n-m}, \forall t \in [0, \tau].
\end{aligned}
\tag{46}
$$

With the error dynamics in Equation (41), we have:

$$
\begin{aligned}
\tilde{x}(iT_s + t) &= e^{A_m t}\tilde{x}(iT_x) + \int_0^t e^{A_m(t-\xi)}B\begin{bmatrix}\hat{\theta}_1^T(iT_s)\\\hat{\theta}_2^T(iT_s)\end{bmatrix}x(iT_s)d\xi \\
&\quad + \int_0^t e^{Am(t-\xi)}B\begin{bmatrix}\hat{\sigma}_1(iT_s)\\\hat{\sigma}_2(iT_s)\end{bmatrix}d\xi - \int_0^t e^{A_m(t-\xi)}B_m\eta_1(iT_s+\xi)d\xi \\
&\quad - \int_0^t e^{A_m(t-\xi)}B_{um}\eta_2(iT_s+\xi)d\xi \\
&= e^{A_m t}\tilde{x}(iT_s) + \int_0^t e^{A_m(t-\xi)}B\begin{bmatrix}\hat{\theta}_1^T(iT_s) & \hat{\sigma}_1(iT_s)\\\hat{\theta}_2^T(iT_s) & \hat{\sigma}_2(iT_s)\end{bmatrix}\Psi(iT_s)d\xi \\
&\quad - \int_0^t e^{A_m(t-\xi)}B_m\eta_1(iT_s+\xi)d\xi - \int_0^t e^{A_m(t-\xi)}B_{um}\eta_2(iT_s+\xi)d\xi.
\end{aligned}
\tag{47}
$$

Let

$$
\begin{aligned}
\zeta_1(iT_s+t) &= e^{A_m t}\tilde{x}(iT_s) + \int_0^t e^{A_m(t-\xi)}B\begin{bmatrix}\hat{\theta}_1^T(iT_s) & \hat{\sigma}_1(iT_s)\\\hat{\theta}_2^T(iT_s) & \hat{\sigma}_2(iT_s)\end{bmatrix}\Psi(iT_s)d\xi, \\
\zeta_2(iT_s+t) &= \int_0^t e^{A_m(t-\xi)}B_m\eta_1(iT_s+\xi)d\xi + \int_0^t e^{A_m(t-\xi)}B_{um}\eta_2(iT_s+\xi)d\xi.
\end{aligned}
\tag{48}
$$

Proving the global boundedness of $\tilde{x}_\tau$ directly presents a challenging task. We can first establish that $x(\tilde{j}T_s)$ ($j$ is the index of the moment) is bounded at all discrete points, and then by examining $x(i\tilde{T_s}+t)$ in Equation (47) for t $\in$ [0, Ts), as $i$ is an arbitrary integer $((i+1)T_s < \tau$ holds) we can determine whether $\tilde{x}_\tau$ is globally bounded.

We now establish that $x(\tilde{j}T_s)$, where $j = 1, 2\ldots$, is bounded at all discrete points, implying that the following equation holds.

$$\|\tilde{x}(jT_s)\|_2 \le 2\zeta(T_s), \ \forall jT_s \le \tau. \tag{49}$$

Clearly, $\|\tilde{x}(0)\|_2 \le \zeta(T_s)$. Furthermore, considering two arbitrarily chosen adjacent discrete points at times $jT_s$ and $(j+1)T_s$, where $(j+1)T_s < \tau$, we have:

$$\tilde{x}[(j+1)T_s] = \zeta_1[(j+1)T_s] - \zeta_2[(j+1)T_s], \tag{50}$$

with

$$
\begin{aligned}
\zeta_1[(j+1)T_s] &= e^{A_m T_s}\tilde{x}(jT_s) + \int_0^{T_s} e^{A_m(T_s-\xi)}B\begin{bmatrix}\hat{\theta}_1^T(jT_s) & \hat{\sigma}_1(jT_s)\\\hat{\theta}_2^T(jT_s) & \hat{\sigma}_2(jT_s)\end{bmatrix}\Psi(jT_s)d\xi, \\
\zeta_2[(j+1)T_s] &= \int_0^{T_s} e^{A_m(T_s-\xi)}B_m\eta_1(jT_s+\xi)d\xi + \int_0^{T_s} e^{A_m(T_s-\xi)}B_{um}\eta_2(jT_s+\xi)d\xi.
\end{aligned}
\tag{51}
$$

By considering the laws of adaptation (Equation (9)), we obtain:

$$\tilde{x}[(j+1)T_s] = h(jT_s) - h[(j+1)T_s]. \tag{52}$$

Let $h(jT_s) = \int_0^{T_s} e^{A_m(T_s-\xi)}B_m\eta_1[(j-1)T_s+\xi]d\xi + \int_0^{T_s} e^{A_m(T_s-\xi)}B_{um}\eta_2[(j-1)T_s+\xi]d\xi$.

Using Equations (20)–(22), we have:

$$\|h(jT_s)\|_2 \leq \kappa_1(T_s)\Delta_1 + \kappa_2(T_s)\Delta_2 = \zeta(T_s), \tag{53}$$

which means that:

$$\|\tilde{x}[(j+1)T_s]\|_2 \leq 2\zeta(T_s), \ \forall (j+1)T_s < \tau. \tag{54}$$

We have now completed the proof that $x(\tilde{j}T_s)$ is bounded at all discrete points. Next, we consider t $\in$ [0, Ts) for all discrete points and prove that $x(iT_s\tilde{} + t)$ is bounded at all times. Then, we can conclude that $\tilde{x}_\tau$ is bounded.

Using the definition of the pseudo-inverse matrix [33], we have $[\Psi(t)]^+ \cdot \Psi \cdot [\Psi(t)]^+ = [\Psi(t)]^+$. Since $\Psi(t)$ is a $(n+1) \times 1$ vector, $[\Psi(t)]^+ \cdot \Psi(t)$ is a constant number. So:

$$[\Psi(t)]^+ \cdot \Psi(t) = 1. \tag{55}$$

Substituting the laws of adaptation (Equation (9)) and Equation (55) into Equation (47) gives:

$$
\begin{aligned}
\|\tilde{x}(iT_s + t)\|_2 &\leq \|e^{A_m t}\tilde{x}(iT_s)\|_2 + \int_0^t e^{A_m(t-\zeta)}B[-\Phi^{-1}(T_s)e^{A_m T_s}\tilde{x}(iT_s) \\
&\quad + \Phi^{-1}(T_s)h(iT_s)][\Psi(iT_s)]^+\Psi(iT_s)d\xi\|_2 + \|\int_0^t e^{A_m(t-\xi)}B_m \\
&\quad \eta_1(iT_s + \xi)d\xi\|_2 + \|\int_0^t e^{A_m(t-\xi)}B_{um}\eta_2(iT_s + \xi)d\xi\|_2 \\
&\leq \|e^{A_m t}\|_2\|\tilde{x}(iT_s)\|_2 + \|\int_0^t e^{A_m(t-\xi)}B\Phi^{-1}(T_s)e^{A_m T_s}d\xi\|_2 \cdot \\
&\quad \|\tilde{x}(iT_s)\|_2 + \|\Phi(t)\Phi^{-1}(T_s)h(iT_s)\|_2 + \|\int_0^t e^{A_m(t-\xi)}B_m d\xi\|_2 \cdot \\
&\quad \|\eta_1(iT_s + \xi)\|_2 + \|\int_0^t e^{A_m(t-\xi)}B_{um}d\xi\|_2\|\eta_2(iT_s + \xi)\|_2 \\
&\leq \bar{\alpha}_1(T_s)2\zeta(T_s) + \bar{\alpha}_2(T_s)2\zeta(T_s) + \zeta(T_s) + \bar{\alpha}_3(T_s)\Delta_1 \\
&\quad + \bar{\alpha}_4(T_s)\Delta_2 \\
&= \gamma_0(T_s).
\end{aligned}
\tag{56}
$$

So, for all $t \in [0, T_s)$, we have $\|\tilde{x}(iT_s + t)\|_2 \leq \overline{\gamma}_0$.

Finally, with the definition of the vector norm, we have:

$$\|\tilde{x}(iT_s + t)\|_{L_\infty} \leq \|\tilde{x}(iT_s + t)\|_2 \leq \overline{\gamma}_0(T_s), \tag{57}$$

which proves Lemma 2.

**Remark 2.** *The boundedness of the error dynamics indicates that the deviation of the state vector between the reference system and the object of control can be limited to within a certain range. Therefore, if the sampling time $T_s$ is properly implemented and the reference system is stable, the object of control remains stable for all times.*

**Lemma 3.** *As the sampling time $T_s$ tends to zero, $\gamma_0(T_s)$ tends to zero, i.e., $\lim_{T_s \to 0} \gamma_0(T_s) = 0$.*

**Proof.** From the definitions of $\kappa_1(T_s)$ and $\kappa_2(T_s)$, we have:

$$\lim_{T_s \to 0} \kappa_1(T_s) = 0, \ \lim_{T_s \to 0} \kappa_2(T_s) = 0. \tag{58}$$

□

Since $\Delta_1$, $\Delta_2$, and $\rho$ are bounded, we have:

$$\lim_{T_s \to 0} \zeta(T_s) = 0. \tag{59}$$

Since $\alpha_1(T_s)$, $\alpha_2(T_s)$, $\alpha_3(T_s)$, $\alpha_4(T_s)$ are continuous, we have:

$$\lim_{T_s \to 0} \overline{\alpha}_1(T_s)(T_s) = 0, \quad \lim_{T_s \to 0} \overline{\alpha}_2(T_s) = 0,$$
$$\lim_{T_s \to 0} \overline{\alpha}_3(T_s)(T_s) = 0, \quad \lim_{T_s \to 0} \overline{\alpha}_4(T_s) = 0. \tag{60}$$

Finally, using the definition of $\gamma_0(T_s)$, we have:

$$\lim_{T_s \to 0} \gamma_0(T_s) = 0. \tag{61}$$

This completes the proof.

**Theorem 1.** *If the sampling time $T_s$ satisfies the condition $\gamma_0(T_s) < \overline{\gamma}_0$ and the $L_1$-PcEx controller satisfies Equation (16), the initial state is $\|x_0\|_\infty \le \rho_0$, and we have:*

$$\|x\|_{L_\infty} \le \rho, \tag{62}$$
$$\|u\|_{L_\infty} \le \rho_u, \tag{63}$$
$$\|\overline{x}\|_{L_\infty} \le \overline{\gamma}_0, \tag{64}$$
$$\|x_{ref} - x\|_{L_\infty} \le \gamma_1, \tag{65}$$
$$\|u_{ref} - u\|_{L_\infty} \le \gamma_2, \tag{66}$$
$$\|y_{ref} - y\|_{L_\infty} \le \|C\|_\infty \gamma_1. \tag{67}$$

**Proof.** In this part, we make a proof by contradiction. $\square$

We assume that the bounds of Equations (65) and (66) do not hold. Since $\|x_{ref}(0) - x(0)\|_\infty = 0 < \gamma_1$, $\|u_{ref}(0) - u(0)\|_\infty = 0 < \gamma_2$ and $x(t)$, $x_{ref}(t)$, $u(t)$, $u_{ref}(t)$ are continuous, there must exist $\tau$ such that:

$$\begin{cases} \|x_{ref}(\tau) - x(\tau)\|_\infty = \gamma_1 \text{ or } \|u_{ref}(\tau) - u(\tau)\|_\infty = \gamma_2 \\ \|x_{ref}(t) - x(t)\|_\infty < \gamma_1, \|u_{ref}(t) - u(t)\|_\infty < \gamma_2, \forall t \in [0, \tau). \end{cases} \tag{68}$$

This implies that at least one of the following equalities holds:

$$\|(x_{ref} - x)_\tau\|_{L_\infty} = \gamma_1,$$
$$\|(u_{ref} - u)_\tau\|_{L_\infty} = \gamma_2. \tag{69}$$

Similarly, according to Assumption 3, we have:

$$\|z_\tau\|_{L_\infty} \le L_z(\|x_{ref\ \tau}\|_{L_\infty} + \gamma_1) + B_z. \tag{70}$$

Lemma 1 implies that:

$$\|x_{ref\ \tau}\|_{L_\infty} \le \rho_r, \ \|u_{ref\ \tau}\|_{L_\infty} \le \rho_{ur}, \tag{71}$$

Using the definitions of Equation (18), we have the following bounds:

$$\|x_\tau\|_{L_\infty} \le \rho_r + \gamma_1 \le \rho,$$
$$\|u_\tau\|_{L_\infty} \le \rho_{ur} + \gamma_2 \le \rho_u. \tag{72}$$

According to Lemma 2, if the sampling time for adaptation $T_s$ is chosen, we have:

$$\|\tilde{x}_\tau\|_{L_\infty} \leq \overline{\gamma}_0. \tag{73}$$

Let $\tilde{\eta}(t) = \tilde{\eta}_1(t) + \tilde{\eta}_{2m}(t)$, where $\tilde{\eta}_1(t) = \hat{\theta}_1^T(t)x(t) + \hat{\sigma}_1(t) - \eta_1(t)$, and $\tilde{\eta}_{2m}(s) = H_m^{-1}(s) H_{um}(s)[\hat{\theta}_2^T(s)x(s) + \hat{\sigma}_2(s) - \eta_2(s)]$.

Using the laws of adaptation and the model of the controlled object, we have:

$$
\begin{aligned}
u(s) &= -C(s)[\eta_1(s) + \eta_{2m}(s) - K_g(s)r(s) - \tilde{\eta}(s)], \\
x(s) &= G_m(s)\eta_1(s) + G_{um}(s)\eta_2(s) - H_{xm}(s)C(s)\tilde{\eta}(s) + H_{xm}(s)C(s)K_g(s)r(s) + x_{in}.
\end{aligned} \tag{74}
$$

Similarly, by using the laws of adaptation and the model of the reference system, we have:

$$
\begin{aligned}
u_{ref}(s) &= -C(s)[\eta_{1ref}(s) + H_m^{-1}(s)H_{um}(s)\eta_{2ref}(s) - K_g(s)r(s)], \\
x_{ref}(s) &= G_m(s)\eta_{1ref}(s) + G_{um}(s)\eta_{2ref}(s) + H_{xm}(s)C(s)K_g(s)r(s) + x_{ref\ in}.
\end{aligned} \tag{75}
$$

So,

$$x_{ref}(s) - x(s) = G_m(s)[\eta_{1ref}(s) - \eta_1(s)] + G_{um}(s)[\eta_{1ref}(s) - \eta_1(s)] + H_{xm}(s)C(s)\tilde{\eta}(s). \tag{76}$$

From the error dynamics (Equation (41)), we have $H_m^{-1}(s)C\tilde{x}(s) = \tilde{\eta}(s)$, and then we obtain:

$$
\begin{aligned}
x_{ref}(s) - x(s) =\ &G_m(s)[\eta_{1ref}(s) - \eta_1(s)] + G_{um}(s)[\eta_{2ref}(s) - \eta_2(s)] \\
&+ H_{xm}(s)C(s)H_m^{-1}(s)C\tilde{x}(s).
\end{aligned} \tag{77}
$$

Therefore, we have:

$$
\begin{aligned}
\|(x_{ref} - x)_\tau\|_{L_\infty} \leq\ &\|G_m(s)\|_{L_1}\|[\eta_{1ref}(s) - \eta_1(s)]_\tau\|_{L_\infty} + \|G_{um}(s)\|_{L_1} \cdot \\
&\|[\eta_{2ref}(s) - \eta_2(s)]_\tau\|_{L_\infty} + \|H_{xm}(s)C(s)H_m^{-1}(s)C\|_{L_1}\|\tilde{x}_\tau\|_{L_\infty}.
\end{aligned} \tag{78}
$$

According to Assumption 3, we have:

$$\|z_\tau\|_{L_\infty} \leq L_z(\rho_r + \gamma_1) + B_z, \tag{79}$$

and hence, we have:

$$
\begin{aligned}
\|X_\tau\|_{L_\infty} &\leq max\{\rho_r + \gamma_1, L_z(\rho_r + \gamma_1) + B_z\} \leq \overline{\rho}_r(\rho_r), \\
\|X_{ref\ \tau}\|_{L_\infty} &\leq max\{\rho_r, L_z\rho_r + B_z\} \leq \overline{\rho}_r(\rho_r).
\end{aligned} \tag{80}
$$

Because $K_{i\overline{\rho}_r(\rho_r)} < L_{i\rho_r}$, we have:

$$
\begin{aligned}
\|(\eta_{iref} - \eta_i)_\tau\|_{L_\infty} &\leq K_{i\overline{\rho}_r(\rho_r)}\|(X_{ref} - X)_\tau\|_{L_\infty} \\
&= K_{i\overline{\rho}_r(\rho_r)}\|(x_{ref} - x)_\tau\|_{L_\infty} \\
&< L_{i\rho_r}\|(x_{ref} - x)_\tau\|_{L_\infty}
\end{aligned} \tag{81}
$$

for $i = 1, 2$.

Therefore, we obtain:

$$
\begin{aligned}
\|(x_{ref} - x)_\tau\|_{L_\infty} \leq\ &\|G_m(s)\|_{L_1}L_{1\rho_r}\|(x_{ref} - x)_\tau\|_{L_\infty} + \|G_{um}(s)\|_{L_1}L_{2\rho_r} \cdot \\
&\|(x_{ref} - x)_\tau\|_{L_\infty}\|H_{xm}(s)C(s)H_m^{-1}(s)C\|_{L_1}\|\tilde{x}_\tau\|_{L_\infty},
\end{aligned} \tag{82}
$$

which, along with the $L_1$-norm condition (Equation (16)) and the definition of $\gamma_1$ (Equation (14)), leads to:

$$\|(x_{ref} - x)_\tau\|_{L_\infty} \leq \gamma_1 - \beta < \gamma_1. \tag{83}$$

Similarly, we have

$$u_{ref}(s) - u(s) \leq -C(s)[\eta_{1ref}(s) - \eta_1(s)] - C(s)H_m^{-1}(s)H_{um}(s) \cdot$$
$$[\eta_{2ref}(s) - \eta_2(s)] + C(s)H_m^{-1}(s)C\tilde{x}(s), \tag{84}$$

Therefore, we have the following bound:

$$\|[u_{ref}(s) - u(s)]_\tau\|_{L_\infty} \leq \|C(s)\|_{L_1}L_{1\rho_r}(\gamma_1 - \beta) + \|C(s)H_m^{-1}(s)H_{um}(s)\|_{L_1} \cdot$$
$$L_{2\rho_r}(\gamma_1 - \beta) + \|C(s)H_m^{-1}(s)C\|_{L_1}\overline{\gamma}_0 \tag{85}$$
$$< \gamma_2.$$

We note that the upper bounds in Equations (83) and (85) contradict the equalities in Equation (69), which proves the bounds in Equations (72), (83), and (85). So, Equations (62), (63), (65), and (66) hold. Moreover, according to Lemma 2, we know that Equation (64) holds.

From the error dynamics in Equation (41), we have:

$$\|y_{ref} - y\|_{L_\infty} \leq \|C\|_\infty \gamma_1, \tag{86}$$

which proves Equation (67).

So, Equations (62)–(67) all hold, which proves Theorem 1.

**Remark 3.** *Theorem 1 states that the tracking error between $y(t)$ and $y_{ref}(t)$, as well as $u(t)$ and $u_{ref}(t)$, is uniformly bounded by an arbitrarily small constant. Thus, an appropriate choice of $T_s$ can realize effective transient and steady-state performance and meet engineering requirements for good tracking, demonstrating the firm stability of $L_1$-PcEx.*

The ability of the $L_1$ adaptive controller to estimate the uncertainties in the adaptation laws and compensate for them in the control signal contributes to its excellent performance and also indicates its great robustness. In an ideal scenario, all uncertainties can be accurately estimated by the adaptation law. However, in reality, estimation errors and delays exist. $L_1$-ExPc demonstrates superior performance in estimating uncertainties compared to $L_1$-Pc due to the smaller upper bounds $\gamma_1$ and $\gamma_2$, indicating the stronger robustness of $L_1$-PcEx. Under different system dynamics, the $L_1$-PcEx controller consistently exhibits the ability to estimate and compensate for uncertainties under different system dynamics, demonstrating its strong adaptability.

## 5. Simulation

As stated in Section 2, we regarded the longitudinal motion of a high-subsonic UAV as the object of control. We used numerous simulations to verify $L_1$-PcEx: simulations of flight feasibility and of algorithmic comparisons as well as of Monte Carlo.

A block diagram of the base controller of the internal loop of the UAV is shown in Figure 2, where this is a proportion differentiation (PD) controller. $\vartheta_{cmd}$ s is the command of the pitch angle, and $K_P$, $K_D$, and $k_g$ are constant gains.

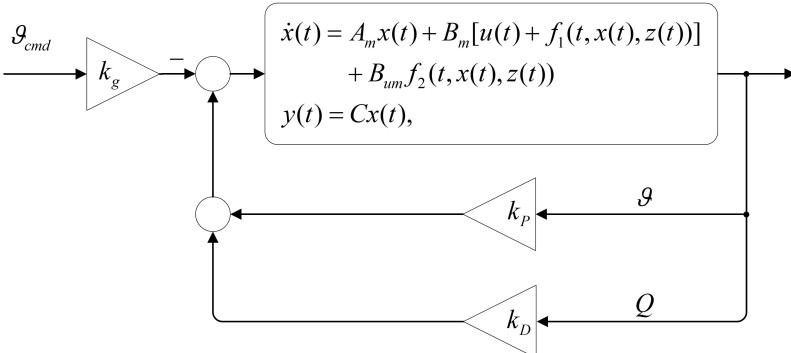

**Figure 2.** Structure of the basic controller.

### 5.1. Flight Feasibility Simulation

This subsection considers the feasibility of $L_1$-PcEx. In the simulations, the command of the pitch angle uses the feedback data from the on-board guidance data of an actual flight. External perturbations were considered.

Figure 3a shows the tracking of the pitch angle using the $L_1$-PcEx controller. The upper half of the figure shows a contrast chart of the actual pitch angle (solid black line) and the command of the pitch angle (dashed orange line). The lower half shows a contrast chart of the estimated pitch angle (dotted red line) and the actual pitch angle (solid black line). Figure 3b shows the effect of tracking the pitch rate (solid black line: pitch rate, dotted red line: the estimated pitch rate).

It is clear that the actual pitch angle accurately tracked its command well and that the estimated state vector accurately tracked its actual value. The 2-norm of pitch error in the simulation satisfied $\|\vartheta_{cmd} - \vartheta\|_2 \leq 65.04$, which decreased by 30.7% compared with that of the actual flight. The mean squared error (MSE) of the pitch was 11.66, and the MSE of the pitch rate was 66.73.

The results verify that $L_1$-PcEx is a feasible flight control system for high-speed UAVs. However, more simulations are needed to be conducted to verify the performance of the $L_1$-PcEx controller; the associated results are in Section 5.2.

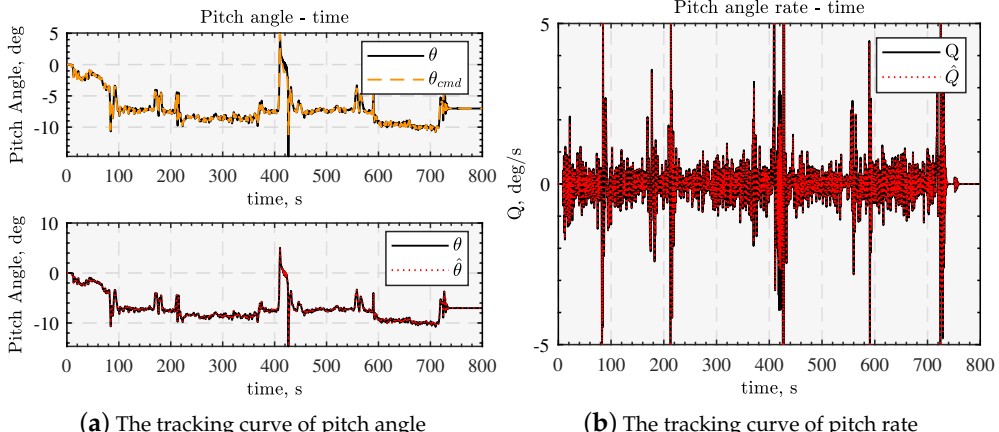

(**a**) The tracking curve of pitch angle

(**b**) The tracking curve of pitch rate

**Figure 3.** Result of flight feasibility simulation.

### 5.2. Comparison in the Presence of Uncertainties

This subsection verifies the performance of the $L_1$-PcEx controller compared with the L1-Pc controller in the presence of uncertainties. Using Ref. [21], we considered the following uncertainties in this part, where $A_\Delta$ represents matched uncertainties, $z(t)$ and $x_z(t)$ represent unmodeled dynamics, and $f_\Delta(t, x(t), z(t))$ represents disturbances (the last

term) and unmatched uncertainties (the other terms). It is easy to prove that, when $x$ is bounded, both $z(t)$ and $f_\Delta(t, x(t), z(t))$ are bounded.

$$f_\Delta(t, x(t), z(t)) = \begin{bmatrix} 3/100x_1 tanh(x_1/2) - \sqrt{|x_1|}/3 - x_2^2/30 - 0.5z \\ -x_2 sechx_2/2 + (1 - e^{-0.3t})/10 + 0.5z \end{bmatrix},$$

$$g(t, x_z(t), x(t)) = [x_{z2}(t); -x_{z1}(t) + 0.8(1 - x_{z1}^2(t)x_{z2}(t))], x_z(0) = [0.5; 0.2],$$

$$g_0(t, x_z(t)) = 0.1(x_{z1}(t) - x_{z2}(t)) + z_u(t),$$

$$z_u(s) = \frac{-s+1}{100s^2 + 8s + 1}[1, -2]x(s),$$

$$A_\Delta = [0.2 \quad -0.2; \quad 0.1 \quad -0.4].$$

(87)

In this subsection, two types of pitch angle commands were used, including square (shown in Figure 4) and sinusoidal signals (shown in Figure 5), to demonstrate that $L_1$-PcEx outperforms $L_1$-Pc under different conditions.

Figure 4a,c show the pitch angle tracking by using the $L_1$-Pc controller and the $L_1$-PcEx controller in the presence of the above uncertainties. The upper halves show a contrast chart of the pitch angle (solid gray line) and the command of the pitch angle (dashed blue line), and the lower halves show a contrast chart of the estimated pitch angle (dotted red line) and the actual pitch angle (solid gray line).

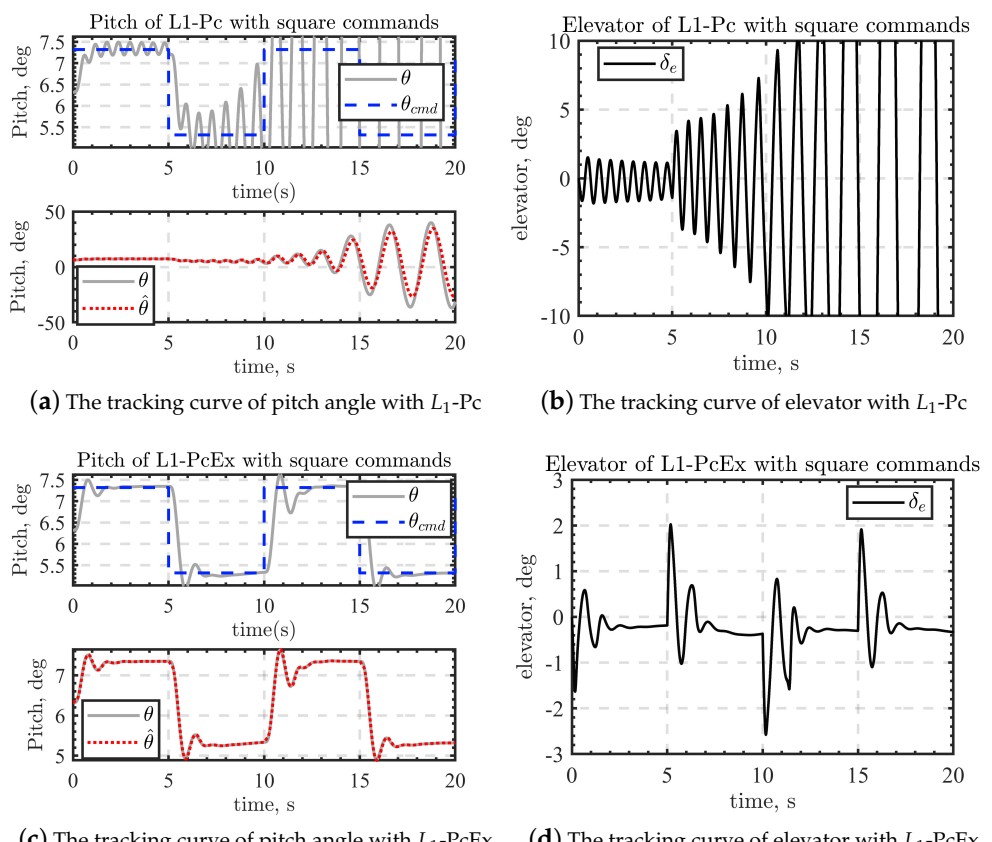

(**a**) The tracking curve of pitch angle with $L_1$-Pc

(**b**) The tracking curve of elevator with $L_1$-Pc

(**c**) The tracking curve of pitch angle with $L_1$-PcEx

(**d**) The tracking curve of elevator with $L_1$-PcEx

**Figure 4.** Performance of $L_1$-PcEx and $L_1$-Pc with square commands.

The convergence time of the pitch angle was less than 2 s for $L_1$-PcEx, and no oscillation occurred. However, it was found that, under the same square wave command, the $L_1$-Pc controller strongly diverges from 6 s onwards. After meticulous comparisons, we found that the controlled deviation of the $L_1$-PcEx controller tends to be smaller than 0.1 degrees, which implies that $L_1$-PcEx meets the accuracy requirements for engineering applications. Meanwhile, the peak time of the $L_1$-PcEx controller was almost the same as that of the $L_1$-Pc controller.

Figure 4b,d show control signals in the presence of uncertainties. The solid black line represents the elevator deflection of the UAV.

As demonstrated in Figure 4b, the elevator signal produced by the $L_1$-Pc controller started to diverge when the pitch angle command changed from 7.6 degrees to 5.6 degrees at 5 s. In contrast, Figure 4d shows that the control signal generated by the $L_1$-PcEx controller consistently converged. This clearly demonstrates that the $L_1$-PcEx controller has a higher adaptive capability than the $L_1$-Pc controller, indicating its potential for superior performance in flight state transitions. Numerically, the MSE of the pitch was 2.33 in $L_1$-PcEx, while it was 516.21 in $L_1$-Pc, which further supports the conclusion mentioned above.

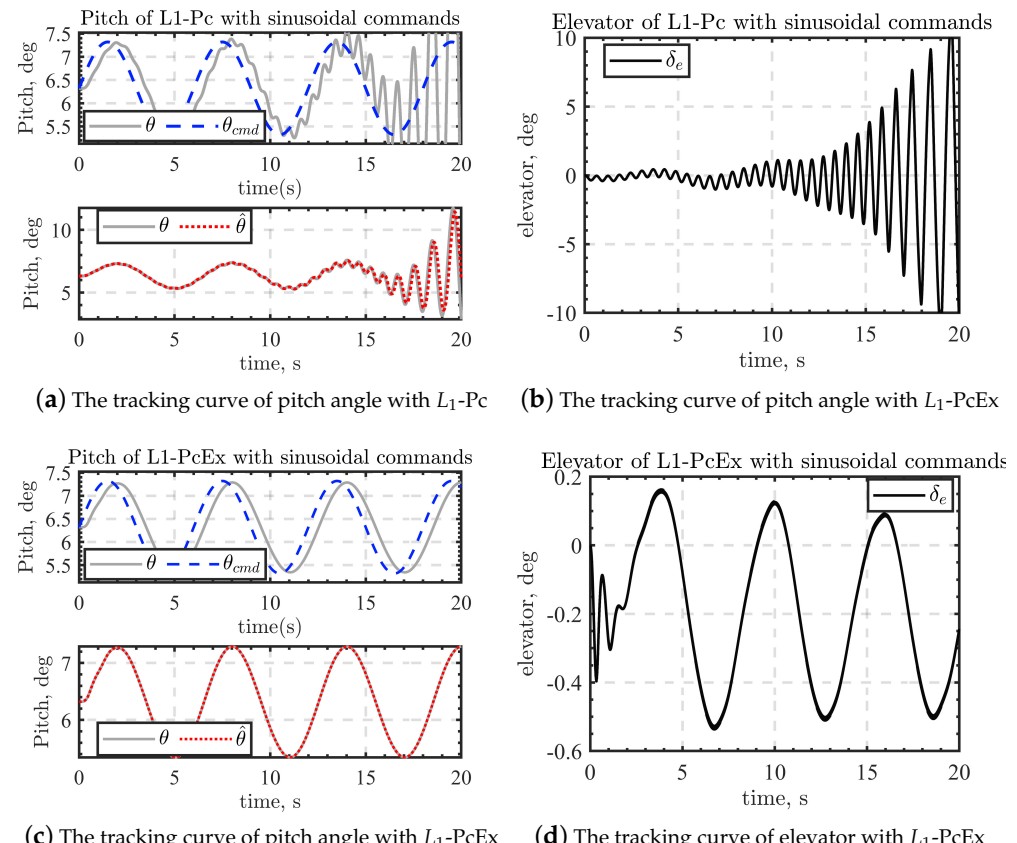

(**a**) The tracking curve of pitch angle with $L_1$-Pc  (**b**) The tracking curve of pitch angle with $L_1$-PcEx

(**c**) The tracking curve of pitch angle with $L_1$-PcEx  (**d**) The tracking curve of elevator with $L_1$-PcEx

**Figure 5.** Performance of $L_1$-PcEx and $L_1$-Pc with sinusoidal commands.

Figure 5 shows simulations performed that are similar to those in Figure 4, but with the pitch angle command replaced by a sine wave signal instead of a square wave signal. The purpose was to evaluate the controller's adaptability and stability, as mentioned earlier. Therefore, the line styles and legends in Figure 5 are consistent with those described earlier and will not be repeated here. The conclusion is also similar: the $L_1$-PcEx controller can adapt to the sinusoidal command within 2–3 s and maintain good tracking, while the $L_1$-Pc controller exceeds the stability margin under the effect of uncertainty and begins to diverge significantly around 13 s. Numerically, the MSE of the pitch was 0.29 in $L_1$-PcEx, while it was 53.62 in $L_1$-Pc.

### 5.3. Monte Carlo Synthesis Verification

Sections 5.2 verified the superior performance of the $L_1$-PcEx controller over the $L_1$-Pc controller. However, to apply the $L_1$-PcEx controller to real flights, further verification is required. We used Monte Carlo simulations to this end.

The scope of deviation for the parameters of the Monte Carlo simulation is shown in Table 1. The lift coefficient, drag coefficient, coefficients of pitching moment, center of

gravity, wind, setting angle, thrust line, inertia, mass, and steerage were related to the model biases. The coefficients of steerage and perturbation were related to the perturbations. All the parameters followed a normal distribution.

**Table 1.** Deviation in Monte Carlo simulation parameters.

| Parameter | Scope |
|---|---|
| Lift coefficient $C_L$ | $\pm 10\%$ |
| Drag coefficient $C_D$ | $\pm 20\%$ |
| Pitching moment coefficient $C_M$ | $\pm 20\%$ |
| Pitching moment coefficient $C_M^{\delta_e}$ | $\pm 30\%$ |
| Pitching moment coefficient $C_M^Q$ | $\pm 50\%$ |
| Center of gravity $x_{CG}$ (m) | $\pm 0.01$ |
| Windx (m/s) | $-10\sim 5$ |
| Setting angle ($^\circ$) | $-1\sim 1$ |
| Thrust line (m) | $\pm 0.05$ |
| Inertia $I_x$ | $\pm 20\%$ |
| Mass (kg) | $\pm 30$ |
| Steerage | $\pm 10\%$ |
| Uncertainties coefficient $k_1 \sim k_7$ [a] | $\pm 2\%$ |
| Uncertainties coefficient $\lambda$ [a] | $\pm 2\%$ |

[a] The coefficient of $f_\Delta(t, x(t), z(t))$.

During the Monte Carlo simulations, we could assess the performance of $L_1$-PcEx during UAV flight state transitions, such as from level flight to descent. However, altitude and speed significantly impact flight characteristics, so it is necessary to introduce outer loop control to maintain them. Thus, we followed the work of Refs. [37,38], and applied the total energy control system (TECS) to the flight control system. The UAV was expected to climb at 5 s, level at 40 s, slide at 75 s, and level again at 110 s. The results of the simulations are shown in Table 2 and Figure 6. It is worth noting that the initial parameters for each Monte Carlo simulation were different, so the given pitch angle command varied.

**Table 2.** Statistical values of Monte Carlo simulation results.

| Parameter | Pitch Angle Error | |
|---|---|---|
| | **Mean Value** | **Sample Variance** |
| Before sliding | $-5.22 \times 10^{-4}$ | $2.48 \times 10^{-7}$ |
| Sliding for 0.5 s | $3.01 \times 10^{-1}$ | $9.48 \times 10^{-5}$ |
| Sliding for 5 s | $-1.52 \times 10^{-1}$ | $7.12 \times 10^{-4}$ |
| Sliding for 10 s | $8.92 \times 10^{-4}$ | $1.13 \times 10^{-5}$ |

Table 2 shows several statistical values for the Monte Carlo simulations, including the mean and sample standard deviation of the slide of the UAV. It is clear that the large error in the pitch angle due to the slide decreased over time and was always less than 0.5 degrees.

Figure 6a–d display the pitch error curve, pitch angle rate curve, adaptive value curve, and elevator curve over time. The solid blue line represents the mean value, and the light blue area represents the sample variance. Figure 6a shows that, although the variance of the error in pitch angle increased to 0.4 degrees when the UAV changed the flight phase and then decreased to less than 0.05 degrees in no more than 10 s, the variance of pitch reached its local minimum value at 43.4 s and decreased by 92.88% within 10 s. As shown in Figure 6b, we also observed an 89.96% reduction in the variance of the pitch rate within 10s. Because TECS gave a different command of pitch angle for each simulation, the adaptive value and the elevator readily stabilized at different values to ensure that the error in the pitch angle tended to zero. Therefore, as seen in Figure 6c,d, the means converged at different states over time while the sample variances remained large.

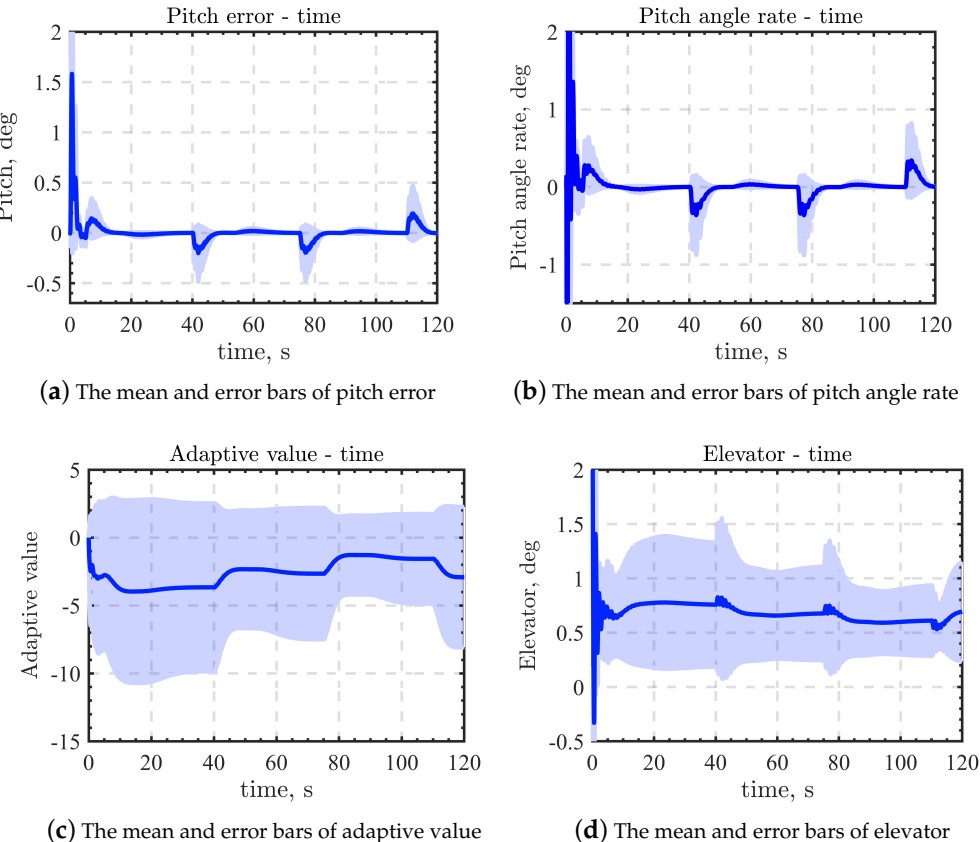

(**a**) The mean and error bars of pitch error

(**b**) The mean and error bars of pitch angle rate

(**c**) The mean and error bars of adaptive value

(**d**) The mean and error bars of elevator

**Figure 6.** Results of Monte Carlo simulations.

## 6. Conclusions

This paper proposed an extended adaptive flight control system for autopilots, highly improving the performance and practicability of the control system. The main motivation for this study was to improve the nonlinear characteristics of the original control system and the extent limitations of this system. We introduced a nonlinear function matrix of state variables to the $L_1$ controller in the adaptation laws, which closely met our expectations. We also employed the actual state vector and its estimate to construct an integrator-like structure and applied it to an $L_1$ controller so that the unexpected transient response produced by the controller can be eliminated. The $L_1$-PcEx controller allowed for a faster adaptation to nonlinear uncertainties compared with the $L_1$-Pc controller. The $L_1$-PcEx controller also delivered a wider margin of stability and higher control accuracy than the $L_1$-Pc. These enhancements were rigorously proven. The results of the simulations verified the superior performance of the $L_1$-PcEx controller to that of the $L_1$-Pc controller.

However, the system we proposed involves the calculation of pseudo-inverse matrices, which places higher demands on CPU performance (approximately an extra 35%). Moreover, the work here is limited by a lack of information on the verification of flights in practice.

In future studies, it would be beneficial to conduct additional verifications to investigate the impact of state vector estimates on the $L_1$ controller. Hardware-in-the-loop simulation (HILS) is an excellent method for validating the effectiveness of algorithms, and we should implement the evaluation of the $L_1$-PcEx controller on HILS using [39] in the future. Additionally, further research can be conducted to enhance the nonlinearity of the system by referring to the Taylor series expansion and investigate the increase in adaptability. Finally, the system proposed in this paper could be further extended and implemented in various other fields, such as self-driving cars, missiles, and spacecraft, to name a few.

**Author Contributions:** Conceptualization, Q.H., Y.F. and L.W.; methodology, Y.F. and B.X.; validation, Q.H. and Y.F.; formal analysis, Q.H., Y.F., L.W. and B.X.; investigation, Q.H. and Y.F.; resources, L.W.; data curation, B.X.; writing—original draft preparation, Q.H.; writing—review and editing, Y.F. and L.W.; visualization, Q.H.; supervision, Y.F.; project administration, L.W.; funding acquisition, L.W. All authors have read and agreed to the published version of the manuscript.

**Funding:** This research was funded by the National Natural Science Foundation of China (Grant No. U21B6003).

**Data Availability Statement:** The data that support the findings of this study are available from the corresponding author upon reasonable request.

**Conflicts of Interest:** The authors declare no conflict of interest.

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
