# Peer review of "A Nonlinear Adaptive Autopilot for Unmanned Aerial Vehicles Based on the Extension of Regression Matrix"

_drones, doi:10.3390/drones7040275_

Round 1

Reviewer 1 Report

1.    Abstract should be improved, for example “enhancing the non-linearity” is not that much clear
2.    The explanation of L1 meaning (line 30) is a bit imprecise
3.    Overall I think a wider state of art should be given while the L1 introduction should be separated in a subsection, also, the need for an integral action is a quite standard problem in control
4.    Model in section 2 should be explained better, also, I have some doubt about matrices dimensionality, can you please check them? Above all it seems some notation is used twice with different meanings
5.    Lines 106-110 seem to be somehow contradictory.
6.    Overall notation is not that much clear, for example the discrete time variable i is never introduced
7.    Assumptions 1-4 should be substantiated (they refer to the specific nested loops architecture), also, assumption 3 notation is not properly introduced
8.    Ts is used twice with different meanings, apparently
9.    Please check equations 21
10.    I’m not that fine with remark 3, moreover, what is the effect of uncertainty in sampling step?
11.    Please explain better remark 4
12.    What is A∆?
13.    The outer loop is used only in 5.3?
14.    In 5.c is theta diverging?
15.    Figures 6.c and 6.d should be replaced with something different, i.e. a figure with mean value and bands expressing the relevant interval
16.    Computation requirements should be clearly expressed and evaluated
17.    “highly improving” is a strong claim, better to express the found improvement in numerical terms
18.    Overall “unexpected tracking error” is not the best label, surely a more common one can be found in literature

Reviewer 2 Report

The manuscript requires major revision before it can be considered for possible publication in this Journal

1.      The authors should clearly write the contribution of this paper. A more detailed discussion regarding the major contribution with regard to the newest existing works should be given to highlight the motivation of this work.

2.      Please include more recent papers 2021-2023 in “Introduction”. There are more recent and relevant references that relate robust control strategies with UAVs:

- Filtered Observer-Based IDA-PBC Control for Trajectory Tracking of a Quadrotor (2021).

- Robust IDA-PBC for under-actuated systems with inertia matrix dependent of the unactuated coordinates: application to a UAV carrying a load (2021).

- Real-time robust tracking control for a quadrotor using monocular vision (2023).

3.      I believe the method developed in this study is simply a combanition and application of existing designs. Similar ideas can also be found in the literature.

4.      More cases with various initial conditions need to be examined to verify the robustness of the proposed methodology.

5.     The stability of the control algorithm must be demonstrated.

6.      Simulations under unmodeled dynamics, parametric uncertainty and disturbances should be obtained.

7.      There are no numerical simulation results, which can demonstrate the superiority of the proposed control law. I suggest comparing the simulations with the results of the recent (2021-2023) related valid references.

8.      The proposed method must be verified via experimental studies. Otherwise there is no way to assess the effectiveness of the proposed method.

9.     The future research direction should be outlined in the conclusion.

Round 2

Reviewer 1 Report

Overall most of my comments have been addressed. I think, however, remark #3 is quite trivial and should be removed.

About figure 6 I still stand my point, even in MC simulations you can use statistical tools for representing data or, even better, a table rather than a superimposition of lines which quickly becomes unreadable, anyway this is fundamentally a choice of yours.

In my opinion, while the numerical evaluation of results, while numerical evaluation is surely hard it should be carried out or, at least, properly introduced to the reader, something e.g. "produced a x% reduction in MSE during a series of tests involving...".

Sorry for being unclear about "figure 5.c", I actually meant the third subfigure of figure 5. This relates, however, to my main residual concern about your paper: the comparison seems to be carried out against a non-working controller. In other words, the L1-PC seems to become unstable during tests, even in a not that much challenging scenario, at least according to figure 5.a.

Reviewer 2 Report

The article is ready to be accepted, as it has met the quality requirements and all the reviewers' comments.

Author Response

Dear reviewer,

Thank you for taking the time to review our manuscript drones-2296337 and for the effort you put into providing us with constructive feedback. We greatly appreciate your contributions to improving the quality of our work. With the help of other reviewers, we have made some revisions to the manuscript, and we welcome your continued assistance in further improving this article if you are interested.

Thank you again for your valuable feedback, and we look forward to hearing from you soon.

Best regards,

Quanwen Hu